# Peripheral leukocyte transcriptomic changes in preweaned Holstein heifer calves with varying stages of Bovine Respiratory Disease

Lindsey C. Makratzakis[1☉‡]*, Joel A. Velasco[2☉‡], Neils C. Stegelmeier[1],
Stephen P. Ficklin[2], Holly R. Hinnant[1], Lily A. Elder[1], Chris M. Mandella[1],
Rachel A. Leeson[1], Carolyn D. Fisher[1], Giovana S. Slanzon[3], Corinna T. Cauchy[1],
Lindsay M. Parrish[1], Kathlyn N. Heaton[1], Craig S. McConnel[1]

1 Department of Veterinary Clinical Sciences, Field Disease Investigation Unit, College of Veterinary Medicine, Washington State University, Pullman, Washington, United States of America, 2 Department of Horticulture, College of Agricultural, Natural, and Resource Sciences, Washington State University, Pullman, Washington, United States of America, 3 Department of Tropical Plant and Soil Sciences, University of Hawaiʻi at Mānoa, Honolulu, Hawaii, United States of America

☉ These authors contributed equally to this work.
‡ These authors share first authorship to this work.
* lindsey.richmond@wsu.edu

## Abstract

Reduction of morbidity and mortality attributable to Bovine Respiratory Disease (BRD) requires a nuanced approach rooted in understanding associated patho-physiological transcriptomic changes. This study aimed to identify 1) transcriptomic changes in peripheral leukocytes indicative of the onset and duration of BRD in pre-weaned Holstein dairy calves, and 2) potential predictive features (i.e., genes) associated with the progression or resolution of BRD underlying disease susceptibility and animal robustness. Neonatal Holstein dairy calves were enrolled across two years (May 2021: n = 60; May 2022 n = 61) on two conventional dairies located in the central region of the state of Washington, United States. Calves were followed for the first 12 weeks of life with sequential thoracic ultrasonographic and clinical assessments. Calves were retrospectively categorized as healthy (2021: n = 8; 2022: n = 15), onset lobar consolidation (2021: n = 16; 2022: n = 15), chronic lobar consolidation (2021: n = 8; 2022: n = 9), or resolved lobar consolidation (2021: n = 7; 2022: n = 11) based upon thoracic ultrasonography. Leukocytes were isolated from whole blood samples via modified Ficoll-Paque separation followed by RNA extraction and sequencing. Differentially expressed genes (DEGs; FDR < 0.05 and |logFC| > 1) were identified via edgeR for each BRD stage compared to healthy: onset (DEG = 163), chronic (DEG = 27) and resolved (DEG = 0). Random forest classification modeling was then conducted to identify gene features with potential to be predictive of disease progression or resolution. Functional enrichment of associated DEGs suggested a shift in resource allocation from primary growth activity to immune response. Identification of key molecular constituents such as interleukins, integrins, serine protease inhibitors

**Data availability statement:** Relevant data are within the paper and its Supporting information files. Data is also available under the project accession number PRJNA1337563 within the National Center for Biotechnology Information Sequence Read Archive. https://dataview.ncbi.nlm.nih.gov/object/PRJNA1337563?reviewer=a6jv0lhokd3h6pnvjaa3i5h8kh Full pipeline analysis will be made available at https://gitlab.com/mcconnel-projects/nifa-ideas-disability-weights'.

**Funding:** Financial support for this project was provided through the United States Department of Agriculture National Institute of Food and Agriculture, Agriculture and Food Research Initiative Competitive Grant no. 2021-68014-34144. The funders had no role in the study design, data collection and analysis, decision to publish, or preparation of the manuscript.

**Competing interests:** The authors have declared that no competing interests exist.

and matrix-associated-remodeling proteins highlight potential intrinsic factors driving differences in disease phenotypes, susceptibility, and animal robustness.

## Introduction

Bovine Respiratory Disease (BRD) results in significant economic loss related to costs associated with treatment, decreased feed efficiency and performance, and mortality. The estimated annual economic loss totals more than $900 million within the United States alone [1,2]. BRD is considered a multifactorial disease complex due to the intricate interactions between the host immune system, viral and bacterial pathogens, and environmental stressors [3,4]. Within the United States, BRD is attributable to 24% of preweaned heifer deaths and 58.9% of weaned heifer deaths [5]. A large cross-sectional study that examined 4,636 calves from 100 dairies within the state of California estimated the prevalence of BRD to be 6.9% (95% CI: 5.7–8.4%) [6]. The range of prevalence (0%−37%) across the 100 farms highlighted the multifactorial nature of BRD dependent upon pathogen etiology, environment, and management practices [6]. Although the estimated case-fatality rate for pre-weaned dairy calves can be relatively low (e.g., 2.4% in a cohort of 11,470 calves), it is important to consider the larger impacts of BRD sequelae, particularly among those with subclinical disease [3].

Research has demonstrated measurable impacts of BRD on various performance indicators such as average daily gain (ADG) and age at first calving (AFC) [7–9]. In one study, heifers diagnosed with BRD had a reduced ADG of 0.1 kg/d [10]. Reduction in ADG can be detrimental due to its impact on sexual maturity, subsequent calf disease incidence and comorbidities [8,11,12]. Heifers diagnosed with BRD as a calf have 2.9 times higher odds of dying and 2.3 times higher odds of herd removal before their first calving compared to BRD-free calves [10]. From an economic perspective, calfhood diseases such as BRD can increase rearing costs up to 6% [13]. In fact, a recent study estimated the cost of BRD in heifers within the first 120 days of age to be $282 [14].

Despite the widespread availability of preventive vaccines and therapeutic antimicrobial drugs labeled for BRD, accurate diagnosis among calves remains a challenge due to the absence of a gold-standard antemortem diagnostic [15,16]. Current diagnosis of BRD is often dependent on recognition of clinical signs, frequently via utilization of a clinical respiratory score (CRS) system [17–19]. In the previously mentioned cross-sectional study, 97% (± 1.7%) of producers utilized a combination of clinical symptom recognition and scoring systems to diagnose BRD [6]. Common clinical signs reported were nasal signs (e.g., dry nose, nasal discharge; 72.0±4.5%), abnormal breathing (56±5.0%), droopy ears or head tilt (44.0±5.0%), and cough (41.0±5.0%) [6]. Despite the improved standardization of clinical assessments associated with CRS systems, the inherent subjectivity with visual assessments results in increased diagnosis of the severe clinical cases and an underdiagnosis of less severe, subclinical cases [10,20].

Recent integration of thoracic ultrasonography (TUS) as a calf-side diagnostic tool provides additional insight into BRD prevalence as it helps overcome the limitations of CRS systems to detect subclinical BRD [20,21]. The sensitivity and specificity of TUS relative to findings at necropsy demonstrated that utilization of TUS to detect BRD-associated pulmonary lesions had a 94% sensitivity and a 100% specificity [21]. However, TUS has a limited ability to detect deep lung lesions due to poor penetration of aerated pulmonary tissue [21]. A recent study examined comparative diagnoses of BRD among preweaned Holstein heifer calves using a standardized TUS technique and CRS system. The study found that the sensitivity of clinical signs to accurately identify lung consolidation was < 40%, with specificity ≥ 92% [22]. The low sensitivity suggests that on-farm diagnoses and treatments of BRD based on clinical signs alone likely underrepresent true disease prevalence, which is in line with the inherent complexities in diagnosing subclinical BRD (i.e., identification of lung consolidation without clinical symptoms of respiratory disease) [23]. Despite the improvement in BRD diagnostic aids (e.g., CRS systems, TUS), the continued high prevalence of BRD underlines fundamental gaps in our understanding of the host response to respiratory infection and our ability to accurately diagnose and prevent disease [4,21,24]. Furthermore, the complexities of the disease pathways call into question how well historical case definitions capture the true nature of the disease [25,26].

The emergence and increased accessibility of transcriptomic technology in medicine has provided an opportunity to overcome some of our knowledge gaps by facilitating the exploration of pathophysiologic nuances between healthy and diseased tissues [27,28]. Utilization of transcriptomic technology in the context of BRD has begun to provide insight into the differences in calf immune response that play a major role in disease severity and susceptibility. Previous research has focused on exploring transcriptomic strategies in experimentally challenged beef cattle, naturally infected beef cattle, and experimentally challenged bull calves revealing significant transcriptomic changes in regulation of genes associated with inflammation, metabolism, growth, and maintenance [29–36]. There is limited literature examining transcriptomic responses among dairy heifers diagnosed with BRD via thoracic ultrasonography within a dairy operation. One important aspect to consider when exploring and applying transcriptomic profiles is that preweaned calves are immunologically distinct from their adult counterparts. Immunophenotyping of 30-day-old healthy, Holstein heifer calves revealed higher proportions of polymorphonuclear (PMN) cells (i.e., neutrophils) and monocytes compared to the immunophenotype profiles of adult Holstein cows as well as an increased expression of CD62L by CD4 + T-cells and B-cells [37]. In fact, factors related to age and immune system development may overshadow disease impacts and influence gene expression patterns in young calves such as the elevated expression of CD62L by lymphocytes [38].

The incorporation of omics approaches with machine learning (ML) strategies has generated a unique opportunity for researchers to interrogate large, complex biological datasets to elucidate intrinsic factors driving health outcomes and disease progression [39–41]. This integration can be viewed as the backbone of precision medicine, which seeks to capture individual variability in gene expression and environmental pressures to address differences in disease phenotypes [42]. Within the livestock industry, omics and ML strategies have become critical tools in livestock improvement programs through the selection of genetically superior animals with increased production, performance, and disease resiliency traits [43,44].

To improve our understanding of BRD pathophysiology in preweaned dairy calves, this study aimed to identify 1) transcriptomic changes in peripheral leukocytes indicative of BRD onset and progression in preweaned Holstein heifer calves, and 2) potential predictive features of BRD progression or resolution to highlight gene features underlying disease susceptibility and animal robustness while addressing the ambiguity of current BRD case definitions.

## Materials and methods

### Ethics statement

The research protocol was reviewed and approved by the Institutional Animal Care and Use Committee of Washington State University (ASAF#6859).

## Experimental design and calf enrollment

This longitudinal study was conducted on two conventional dairies located in the central region of the state of Washington, United States, from May 2021 to August 2021, and May 2022 to August 2022. Farm A maintained approximately 11,000 lactating Holstein cows, and Farm B maintained approximately 2,250 lactating Holstein cows. Convenience cohorts of neonatal Holstein heifers were enrolled based on age and adjacent housing. Cohorts were enrolled on a single day per farm at approximately seven (± 3) days of age during both the May 2021 to August 2021, and May 2022 to August 2022 enrollment periods. In 2021, 30 calves were enrolled at each of the two farms, and in 2022, 31 calves were enrolled on Farm A, and 30 were enrolled on Farm B. Overall, a total of 121 neonatal calves received 10 weekly clinical assessments.

## Calf management

Calves were removed from their dam within approximately 30–60 minutes of birth, had their naval dipped in iodine, and received their identification ear tags. Within 1–2 hours of birth, each calf received one gallon of colostrum (Brix refractometer ≥ 22%) via oral intubation followed by a second gallon of colostrum (Brix refractometer < 22%) 8–12 hours later. Fresh water was freely available to each calf and was refreshed daily. Calves on Farm A were bucket fed pasteurized hospital milk and unpasteurized bulk tank milk mixed with milk replacer as needed to achieve total milk solids of 13–13.5%. They were provided three quarts (2.8 liters) of the milk mixture twice a day from 1–20 days of age, 2.5 quarts (2.4 liters) twice a day from 21–39 days of age, and two quarts (1.9 liters) twice a day from 40–55 days of age at which point they were weaned. A mixed ration of calf pellets, rolled corn, molasses, and small percentage of chopped alfalfa was offered from the first day of age. Calves on Farm B were bucket fed three quarts (2.8 liters) twice daily of pasteurized hospital and bulk tank milk mixed with milk replacer to achieve a total milk solids of 14%. Weaning was abrupt at 60 days of age. Calf grain and alfalfa pellets were available from the first day of age with grain fed to a maximum of 3.0 lb. (1.4 kg) and alfalfa pellets fed ad libitum.

Calves on both dairies were housed in pens that were bedded with straw and had removable solid walls within an open-sided, covered unit housing 60 calves with 30 per side. The 2021 cohort on Farm A was housed in pairs starting in Week 9 by removing the wall between pairs, whereas the 2022 cohort on Farm A was paired from the outset of the study. Farm B calves were housed in individual pens throughout both years. Farm A calves received a single, intranasal dose of Inforce3® (Zoetis Animal Health, Parsippany, NJ, USA) at four weeks of age. Farm B calves received a single, intranasal dose of Nasalgen3® (Merck Animal Health, Kenilworth, NJ, USA) at two weeks of age.

## Clinical assessments and sampling

At enrollment, transfer of passive immunity (TPI) was evaluated through measurement of calf total serum protein and categorized based on industry consensus for passive immunity (e.g., poor: < 5.1 g/dl serum total protein; fair: 5.1–5.7 g/dl serum total protein; good: 5.8–6.1 g/dl serum total protein; excellent: ≥ 6.2) [45]. Weekly clinical assessments were conducted by our research team using a standardized scoring system for clinical respiratory symptoms (CRS) alongside thoracic ultrasonography (TUS) for assessment of pulmonary consolidation. Clinical assessments were performed using the standardized Wisconsin Calf Health Scoring Chart [19]. Parameters (nose, ears, eyes, cough, navel, joints, feces and rectal temperature) were assessed on a 0–3-point scale to reflect clinical sign severity (e.g., 0 = normal, 1 = mild, 2 = moderate, 3 = severe) [19].

Diagnosis of pulmonary pathology was reliant on calf-side ultrasonography utilizing the calf scanning technique previously described by Ollivett and Buczinksi [46]. Examination of the lung fields involved imaging the tenth through first intercostal spaces from the dorsal aspect to the ventral aspect on both the right and left side of the calf. The seven peripheral lung lobes were each scored using a 0–5-point scale as follows: 0 = visually healthy lungs, 1 = diffuse comet tailing, 2 = lobular consolidation of ≥ 1 cm but not full thickness, 3 = full thickness lobar consolidation of one lung lobe, 4 = full thickness

lobar consolidation of two lung lobes, and 5 = full thickness consolidation of three or more lung lobes [46]. Clinical assessments and TUS evaluations were performed under the supervision of the Principal Investigator, McConnel, including secondary review of recorded TUS video for all identified lung lesions. TUS was conducted using IBEX® EVO® II and IBEX® PRO/r ultrasounds both equipped with a L7HD linear probe for large animal exams (E.I. Medical Imaging, Loveland, CO, USA).

Calves enrolled during the May 2021 to August 2021 enrollment period were evaluated for BRD using TUS commencing during the second week of enrollment and conducted weekly thereafter to inform disease status as either healthy, onset-consolidation, chronic-consolidation, or resolved consolidation. Calves enrolled during the May 2022 to August 2022 enrollment period received an initial TUS scan at week 3 of age and were scanned every other week thereafter (e.g., weeks 5, 7, 9, 11) unless identified with lung pathology which resulted in a follow up TUS evaluation on the subsequent even week (e.g., weeks 4, 6, 8, 10). Additionally, calves with a rectal temperature > 103.0°F (39.4°C) on an even week's clinical assessment were evaluated with TUS regardless of the previous week's results.

Across the 12-week evaluation period, up to six blood samples were collected starting at enrollment (i.e., week 1) for the evaluation of complete blood counts, serum blood chemistry, and leukocyte isolation. Approximately 10 mL of blood was collected from the jugular vein into Covidien Monoject coated EDTA evacuated tubes with lavender stopper tops (Fisher Scientific, Waltham, MA, USA) and into Covidien Monoject silicone-coated evacuated tubes with red stopper top (Fisher Scientific, Waltham, MA, USA). All whole blood sample tubes were inverted multiple times prior to being placed on top of ice for transport. Samples were refrigerated for up to 24 hours at 41.0°F (5.0°C) until processing.

## Sample size

Sample size logistics and requirements were determined based on historical precedent among the two participating farms, previous estimates for RNA sequencing data [47], and our own experience with peripheral leukocyte transcriptomics [48]. Participant farm treatment data indicated an expected minimum 15% prevalence of clinical respiratory disease among calves through 12 weeks of age. This equated to no fewer than 18 calves with clinical BRD out of 120 enrollees. Transcriptomic sample size requirements based on a negative binomial distribution suited for optimal statistical design using RNA-Seq at a given significance level of 0.05, power of 80%, a conservative 2-fold estimate of effect, a 30 million read depth of sequencing for a given transcript, and a conservative 0.35 coefficient of variation (CV) for RNA-seq counts within comparative groups suggested a minimum of 5 samples per group (diseased versus healthy) [47].

## Calf selection for transcriptomic analysis

Calves were retrospectively assigned to groups based on case definitions derived from TUS scoring, farm treatment records, and the clinical findings obtained during their ten weekly clinical assessments. Each enrolled calf was evaluated in the context of the following case definitions: *Healthy* (i.e., no evidence of lung pathology via TUS, no evidence of BRD via CRS, no evidence of scours, temperature < 103.0°F (39.4°C)); *Onset* (i.e., initial identification of lobar consolidation based on TUS ≥ 3), *Chronic* (i.e., two or more weeks of TUS ≥ 3), and *Resolved* (i.e., no evidence of lobar pathology via TUS < 3 and no evidence of CRS following a diagnosis of *Onset* or *Chronic*). From the 121 neonatal calves enrolled across the two years, 65 calves were selected based on the case definitions above (S1 Table). Note that calves that were included with the onset group were evaluated for subsequent inclusion in the chronic or resolved categories if they adhered to the respective case definitions. Similarly, calves included in the chronic group were evaluated for subsequent inclusion in the resolved category. Thirty healthy samples (*2021*: week 5, n = 5; week 7, n = 8; week 9, n = 3; *2022*: week 5, n = 6; week 7, n = 4; week 9, n = 4) were used for age matched comparisons with calves at the onset of BRD (*2021*: week 5, n = 6; week 7, n = 10; *2022*: week 5, n = 6; week 7, n = 5; week 9, n = 4), with chronic disease (*2021*: week 7, n = 3; week 9, n = 5; *2022*: week 7, n = 5; week 9, n = 4), or following resolution of disease (*2021*: week 7, n = 3; week 9, n = 4; *2022*: week 7, n = 5; week 9, n = 6).

## Leukocyte RNA extraction and gene expression analysis

Leukocyte isolation from whole blood samples was completed utilizing a modified Ficoll-Paque separation technique with a short erythrocyte lysis stage. A sterile serological pipette was used to transfer 7 mL of whole blood into a sterile 50 mL Nunc™ Conical Sterile Polypropylene Centrifuge Tube (Fisher Scientific, Waltham, MA, USA) that contained 7 mL dPBS(-) [49]. The dPBS(-) diluted whole blood mixture was layered over 10 mL of Ficoll-Paque Premium 1.084 media (GE Healthcare Life Sciences, Marlborough, MA, USA). The conical tubes were then centrifuged for 30 minutes at 400 X $g$ at 68.0°F (20.0°C), and the plasma layer was removed using a sterile pipette. An additional sterile pipette was used to transfer the cellular interface into a new, sterile 50 mL Nunc™ Conical Sterile Polypropylene Centrifuge Tube (Fisher Scientific, Waltham, MA, USA). The cellular interface layer consisting of mononuclear cells, a layer of Ficoll-Paque, and the uppermost portion of the erythrocyte layer that contained polymorphonuclear neutrophils (i.e., granulocytes) was diluted 1:1 with dPBS(-) before it was centrifuged for 10 minutes at 400 X $g$. The resulting supernatant was discarded, and the erythrocyte contamination was removed utilizing a sterile pipette by performing and repeating a lysis step with 5 mL of hypotonic lysis buffer and 5 mL of re-equilibration buffer [50].

The subsequent clarified pellet was then resuspended in 1.2 mL of QIAzol from a miRNeasy kit (Qiagen, Hilden, Germany). The resuspended pellet mixture was transferred into a 2 mL sterile microcentrifuge tube and was homogenized via vortexing for 1 minute before being stored at −112.0°F (−80.0°C) until further processing. The QIAzol/cell mixture was held at −112.0°F (−80.0°C) until the RNA extraction process was completed using the miRNeasy mini kit protocol under the directions for purification of total RNA from animal cells. The samples were thawed to room temperature, where 0.025 mL of proteinase K (4 mg/mL) was added to the thawed QIAzol/cell mixture. This was incubated at ambient temperature for 10 minutes. Appropriate adjustments were made to reagents to meet requirements for volume ratios prior to completing the remaining steps as directed by the miRNeasy kit protocol [51]. Genomic DNA was removed utilizing the DNA clean-up & Concentrator-25 kit (D4033) (Zymo, Irvine, CA, USA). The total quantity of RNA was validated using a NanoDrop 1000 Spectrophotometer (Fisher Scientific, Waltham, MA, USA). The RNA samples were transported on dry ice for further processing by the Novogene Corporation Inc. (Sacramento, CA, USA) for mRNA sequencing and bioinformatics.

## mRNA sequencing

RNA samples were submitted to Novogene Corporation Inc. (Sacramento, CA, USA) where they underwent mRNA sequencing for expression analysis. Before library construction, quality control reports were generated for each sample using the Agilent 2100 Bioanalyzer (Agilent Technologies Inc., Santa Clara, CA, USA). Samples required an RNA integrity number value of ≥ 6.8 to be selected for library preparation [52]. RNA integrity number values less than 6.8 were re-extracted and submitted again. Expression analysis was performed using Illumina Novaseq and Hiseq platforms with a paired end 150 bp sequencing strategy (Novogene, Sacramento, CA, USA).

RNA integrity and purity parameters were evaluated for the 96 representative samples of BRD progression categories (e.g., *Healthy, Onset, Chronic, Resolved*) submitted to Novogene Corporation Inc for RNA sequencing (S2 Table). The RNA integrity number (RIN) ranged from 2.3–9.0 (mean: 6.3±1.4; median: 6.4) with an average pure RNA quantity concentration of 54.1±32.11 ng/μL (range: 11.8–141.7 ng/μL). The threshold for a pass was established as an RIN>4.0. Eight values were considered as failed per Novogene Corporation Inc thresholds (S1 Table). However, further analysis of these eight samples using MultiQC v1.11, found no indications of poor data nor poor read quality based on alignment statistics (S1-S2 Fig) [53]. These eight samples were chosen to be included in the library preparation.

## Gene expression quantification

The subsequent gene expression levels from the leukocyte RNA-seq were quantified using GEMmaker, an nf-core compatible Nextflow workflow, that supports read trimming, multiple sequence aligners and pseudoaligners, expression quantification, and quality checks against the NCBI *Bos taurus* genome (Assembly: ARS-UCD1.3) [54–56]. GEMmaker

(default settings) was instructed to use the Kallisto pseudoaligner [54]. Gene expression matrices (GEMs) generated by GEMmaker were organized to have columns represent individual samples, rows to represent the genes, and cells were populated with the associated expression level counts. The GEM generated by GEMmaker included 67,766 genes (total genes identified in the dataset) across the 53 samples. Illumina 16s mRNA sequence data was deposited into the National Center for Biotechnology Information database under project accession number PRJNA1337563.

The resulting GEM was subdivided into the respective disease groups for analysis: *Healthy* (2021: n = 8; 2022: n = 14) vs *Onset* (2021: n = 16; 2022: n = 15); *Healthy* (2021: n = 8; 2022: n = 14) vs *Chronic* (n = 8; 2022: n = 9); and *Healthy* (2021: n = 8; 2022: n = 14) vs *Resolved* (n = 7; 2022: n = 11). Filtering and normalization were conducted on individual comparison groups by year, then merged using the outer merge method. The resulting three GEMs were reduced to 40,351, 38,260, and 37,888 genes, respectively. The associated quality control figures and full analysis pipeline are available at https://gitlab.com/mcconnel-projects/nifa-ideas-disability-weights.

## Differential gene expression analysis

To better understand the host response, differential gene expression was conducted to examine respiratory disease progression impacts on gene expression profiles of the host. Differential gene expression (DGE) analysis was performed using the edgeR package [57]. Differential gene expression analysis was used to identify genes that were either up- or downregulated with respect to healthy calves (*Healthy* vs *Onset*, *Chronic*, or *Resolved*). To account for potential year and farm effects, both were included as covariates in the model. Selection of differentially expressed genes (DEGs) was based on a false discovery rate (FDR) < 0.05 and an absolute $\log_2$ fold change (logFC) greater than 1. Visual assessment of DEG similarity and dissimilarity was performed utilizing MDS plots. Heat maps of DEGs were generated using hierarchical clustering of samples based on expression levels of genes for each BRD progression category.

## Random forest analysis

Though DGE analysis is a standard protocol for identifying genes that differ in expression between conditions, we were also interested in identifying genes whose expression are predictive and can serve as biomarkers of calf health status. To this end, we employed a Random Forest (RF) supervised machine learning approach. Using the merged (2021 and 2022) log2 transformed TMM condition comparison GEMS (*Healthy* vs *Onset*, *Chronic*, or *Resolved*) as inputs, BorutaPy (of the scikit-learn Python package) was used to reduce overfitting by keeping genes that perform better in comparison to the random shuffling of the data [58,59]. Calf health status was used as the dependent (or target) variable, and the gene expression values were the independent variables. BorutaPy parameters required each gene to perform 95% better than random within 2000 iterations (parameters: max_iter: 2000, perc = 95). Genes that were selected by Boruta were then used for hyper-parameter tuning via cross validation, using scikit-learn's GridSearchCV function (hyperparameter grid: n_estimators: [50, 100, 150, 200], max_depth: [None, 4, 5, 6], min_samples_leaf: [1, 4, 7, 10], min_impurity_decrease: [0.0, 0.01, 0.001, 0.0001], "bootstrap": [True], oob_score: [True, False], max_samples: [None, 0.25, 0.5, 0.75], ccp_alpha: [0.0, 0.005, 0.015, 0.025]), using neg_root_mean_squared_error for scoring. The identified hyperparameters then served as settings for the random forest classifier models in a second round of Boruta selection (parameters: max_iter: 2000, perc = 95, estimator = RandomForestClasssifier(optimized hyperparameters)). Throughout this paper we refer to genes selected using RF models as (RFGs). The resulting RFGs were then filtered using their associated feature importance scores (FI > 0.0). Overlapping DEGs and RFGs were identified per condition comparison across both years of study.

## Functional enrichment analysis

Functional enrichment of the DEGs was conducted for the DEGs to provide initial insight into the underlying molecular mechanisms and biological pathways associated with BRD. To identify the up and downregulated function underlying the identified DEGs, functional enrichment analysis was performed using FUNC-E v2.0.1 (default parameters) [60]. Functional

enrichment analysis utilized Gene Ontology (GO) terms across all three domain levels (i.e., Biological Process, Molecular Function, Cellular Component) as well as InterPro (IPR) terms identified using InterProScan v5.36 (default parameters), its accompanying databases, and the PANTHER 14.1 databases [61–63]. Kyoto Encyclopedia of Genes and Genomes (KEGG) pathways and ortholog terms were assigned using the KEGG Automatic Annotation Service (KAAS) [64,65].

## Results

### Differentially expressed genes

Gene expression analysis for the *Healthy* vs *Onset* comparison identified 163 (148 upregulated, 15 downregulated) DEGs (FDR < 0.05 and |logFC| > 1) in calves defined as *Onset* compared to calves defined as *Healthy* (Fig 1, S3 Table). For the *Healthy* vs *Chronic* comparison, 27 (25 upregulated, 2 downregulated) DEGs (FDR < 0.05 and |logFC| > 1) were identified in calves defined as *Chronic* compared to calves defined as *Healthy* (Fig 2, S4 Table). Lastly, for the *Healthy* vs *Resolved* comparison, no DEGs were identified utilizing the FDR < 0.05 and |logFC| > 1 thresholds (Fig 3). Additional thresholds such as FDR < 0.1 and |logFC| > 0.05 were also considered but did not yield any significant DEGs.

### Identification of potential predictive features

Following the identification of DEGs, a second parallel analysis was conducted to identify genes with a predictive capacity for the respective disease state. Random forest classification modeling was performed across samples to gain additional perspective on both noncoding and coding genes that are potentially predictive of calves at *Healthy*, *Onset*, *Chronic*, or *Resolved* states. The RFGs facilitated the identification of genes that may not necessarily have been differentially expressed due to the model selection criteria.

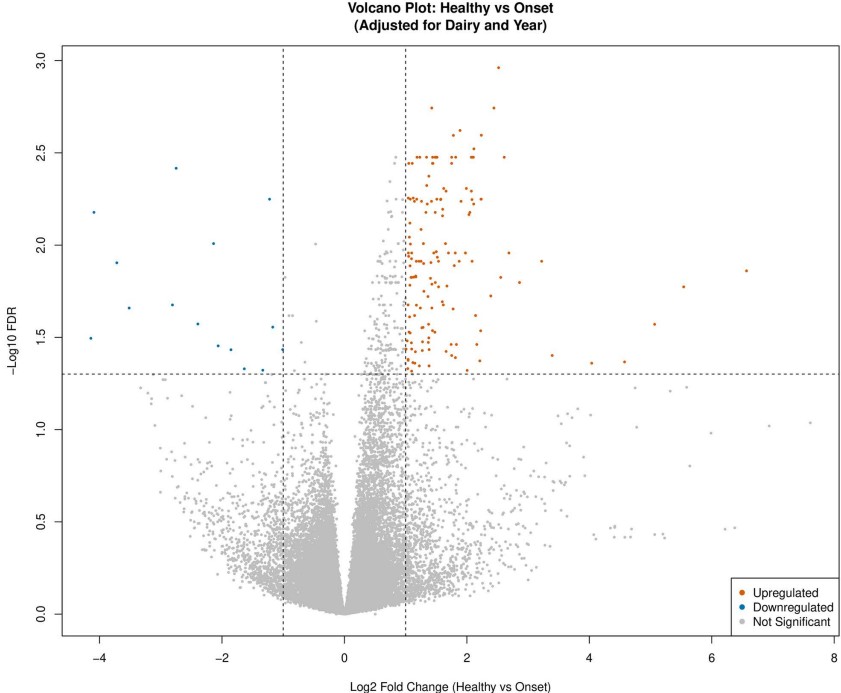

**Fig 1. Volcano plot for *Healthy* vs *Onset* differentially expressed genes (DEGs: n = 163; FDR <0.05 and |logFC| > 1) adjusted for dairy and year.** Upregulated genes are depicted in orange and downregulated genes are highlighted in blue. Non-significant genes are colored grey.

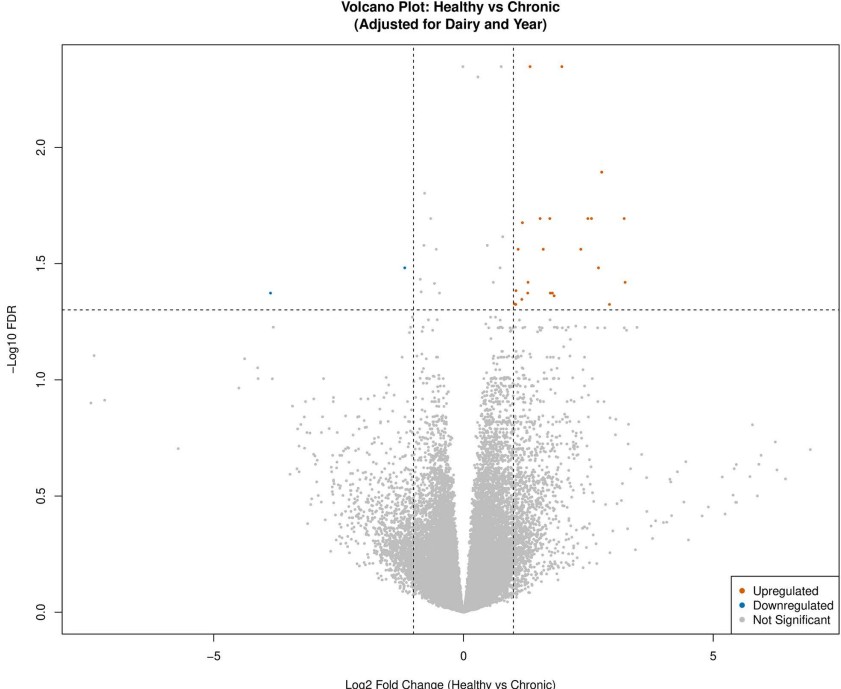

**Fig 2. Volcano plot for *Healthy* vs *Chronic* differentially expressed genes (DEGs: n = 27; FDR < 0.05 and |logFC| > 1) adjusted for dairy and year.** Upregulated genes are depicted in orange and downregulated genes are highlighted in blue. Non-significant genes are colored grey.

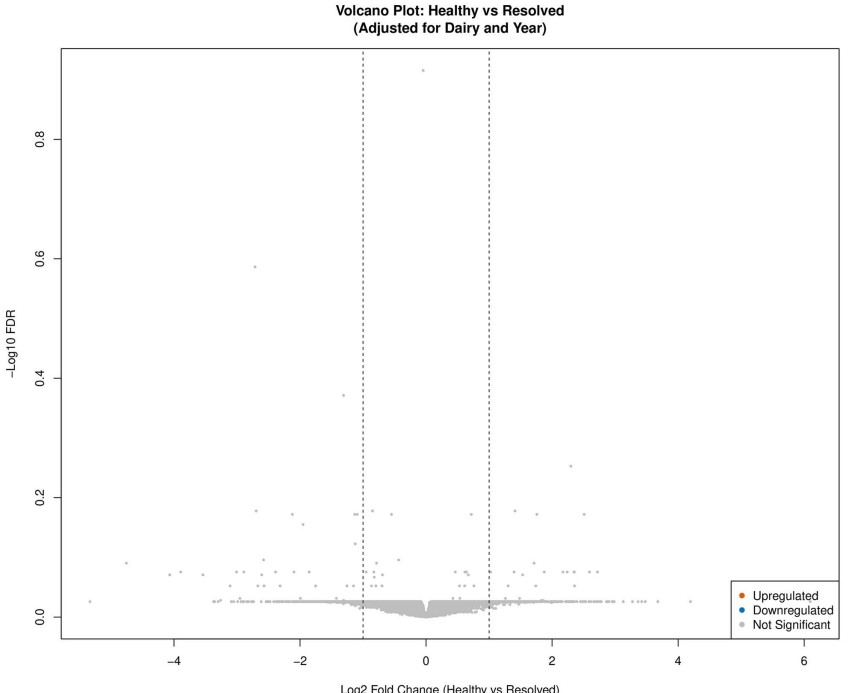

**Fig 3. Volcano plot for *Healthy* vs *Resolved* differentially expressed genes (DEGs: n = 0; FDR < 0.05 and |logFC| > 1) adjusted for dairy and year. Upregulated genes are depicted in orange and downregulated genes are highlighted in blue.** Non-significant genes are colored grey. No significant differentially expressed genes were identified for *Healthy* vs *Resolved*.

Random forest classification modeling for *Healthy* vs *Onset* identified 51random forest gene features with predictive capacity for onset of BRD (S5 Table). The top features identified for *Healthy* vs *Onset* included centriolin (CNTRL), adhesion G protein-coupled receptor E1 (LOC508459), membrane-spanning 4-domains subfamily A member 15 (MS4A15), ankyrin repeat domain-contain protein 33B (ANKRD33B), and flotillin-2 (FLOT2). For *Healthy* vs *Chronic*, 38 RFGs were identified (S6 Table). The top features identified for *Healthy* vs *Chronic* were ATP-binding cassette sub-family A member 9 (ABCA9) and phosphatidylinositol-3, 5-bisphosphate 3-phosphate (MTMR14). For *Healthy* vs *Resolved*, 57 RFGs were identified (S7 Table). The top feature identified for *Healthy* vs *Resolved* was matrix-remodeling-associated protein 5 (MXRA5).

## Gene overlap

Genes identified as potentially predictive (i.e., RFGs) were then compared against the list of DEGs to identify shared genes. For *Healthy* vs *Onset*, six shared genes were identified (Table 1, S3 Fig). Comparatively, for *Healthy* vs *Chronic*, one shared gene was identified (Table 1). Cyclin-dependent kinase 4 inhibitor B (CDKN2B) was identified as differentially expressed and predictive for both *Healthy* vs *Onset* and *Healthy* vs *Chronic* (Table 1, S4 Fig).

## Functional enrichment of differentially expressed genes

**DEG functional enrichment.** Functional enrichment of the 163 DEGs for *Healthy* vs *Onset* resulted in the identification of 225 functionally enriched terms within eight cluster modules of which 179 were upregulated, and 46 were downregulated (S8 Table). Visualization of top enriched terms per module is appreciable in Fig 4. Functional enrichment for the 27 DEGs associated with *Healthy* vs *Chronic* resulted in the identification of 55 functionally enriched terms (9 upregulated, 46 downregulated) within five cluster modules (S9 Table). Top enriched terms can be visualized in Fig 5.

The summary dot plot shows the top enriched terms generated using a Benjamini p-value. The x-axis depicts the gene ratio, the y-axis lists the GO term names, the color intensity signifies FDR significance, and the dot size designates the number of genes in the term.

The summary dot plot shows the top enriched terms generated using a Benjamini p-value. The x-axis depicts the gene ratio, the y-axis lists the GO term names, the color intensity signifies FDR significance, and the dot size designates the number of genes in the term.

**Table 1. Summary characteristics of DEG and RFG feature overlap for BRD comparison groups.**

| Comparison | Gene Symbol | Description | logFC | logCPM | *p*-value | FDR | Importance Score |
|---|---|---|---|---|---|---|---|
| *Healthy* vs *Onset* | LOC104973586 | Bos taurus uncharacterized | 2.61 | 4.21 | 1.24E-06 | 0.003 | 0.034 |
| | CDKN2B | Cyclin-dependent kinase 4 inhibitor B | 1.34 | 2.12 | 3.65E-06 | 0.005 | 0.034 |
| | ITGB5 | Integrin beta-5 precursor | −1.22 | 1.99 | 6.26E-06 | 0.006 | 0.027 |
| | LOC508459 | Adhesion G protein-coupled receptor E1 isoform X14 | 1.57 | 6.64 | 6.70E-06 | 0.006 | 0.027 |
| | LOC508459 | Adhesion G protein-coupled receptor E1 isoform X16 | 1.70 | 4.16 | 2.33E-05 | 0.011 | 0.027 |
| | IL21R | Interleukin-21 receptor isoform X1 | 2.39 | 4.28 | 7.46E-05 | 0.019 | 0.013 |
| | | | | | | | |
| *Healthy* vs *Chronic* | CDKN2B | Cyclin-dependent kinase 4 inhibitor B | 1.29 | 2.06 | 2.78E-05 | 0.038 | 0.001 |

LogFC, Log Fold Change; LogCPM, Log Counts Per Million; FDR, False Discovery Rate.

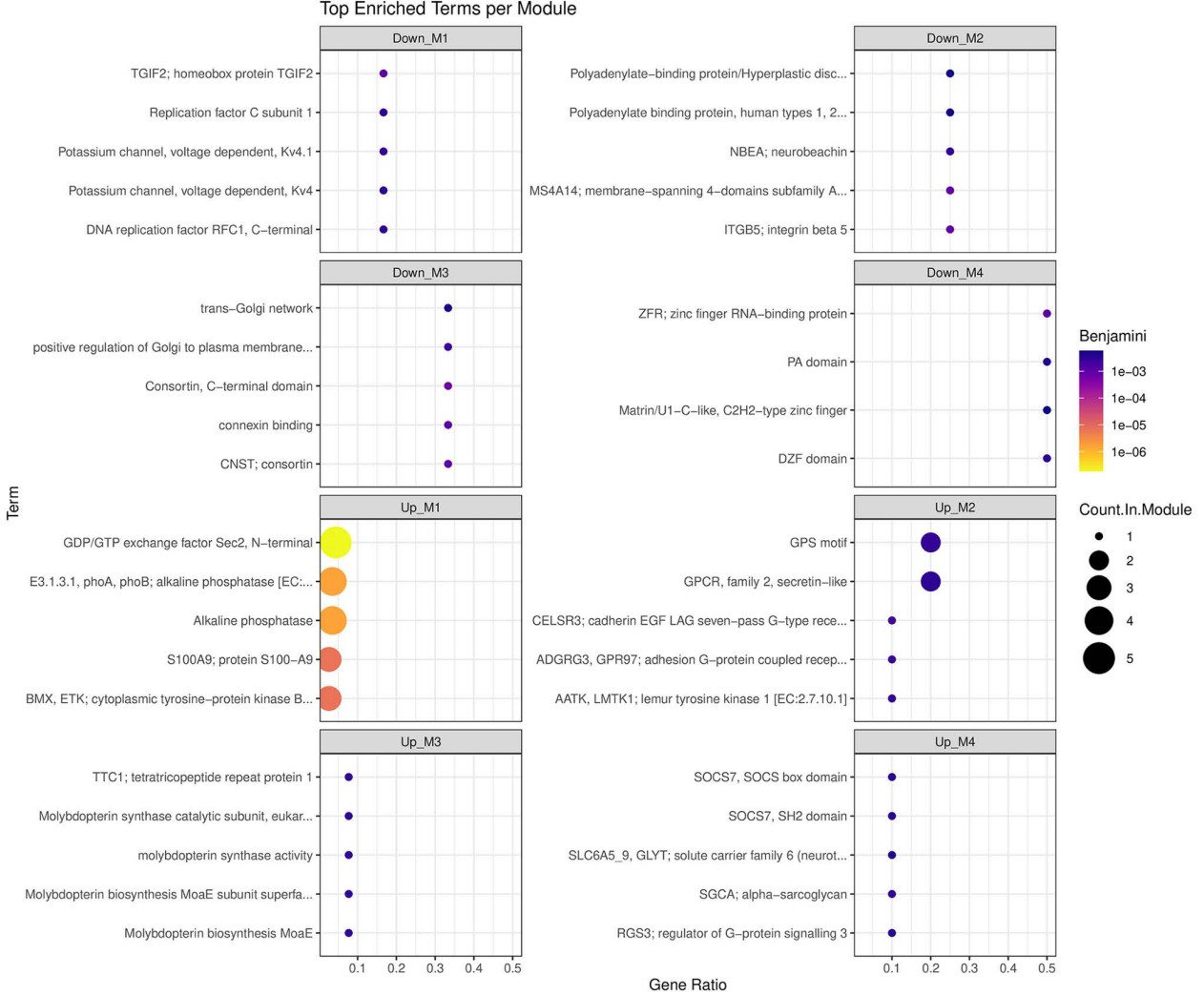

**Fig 4. Summary dot plot of top enriched functional terms for *Healthy* vs *Onset*.**

## Discussion

To the authors' knowledge this is the first prospective study designed to investigate the relative changes in the peripheral leukocyte transcriptome of preweaned Holstein dairy calves diagnosed with varying stages of BRD as determined by thoracic ultrasonography. Differential gene expression is an important tool for elucidating the molecular differences between healthy and diseased animals. Utilization of transcriptomics provides insight into gene regulation and response mechanisms in the context of the environment, which can facilitate the identification of potential biomarkers indicative of diagnosis, prognosis, or treatment efficacy [66]. Moreover, in the context of peripheral leukocytes, interrogation of the peripheral leukocyte transcriptome provides a unique opportunity to gain a functional insight into the host responses at the onset, progression, and resolution of disease.

The onset of respiratory disease is marked by various pathophysiologic changes that begin to alter pulmonary function through inflammation, remodeling of the airways, increased airway resistance, goblet cell hypersecretion, ciliary dysfunction, and gas exchange abnormalities [67,68]. Examination of the peripheral leukocyte transcriptome of *Healthy* vs *Onset*

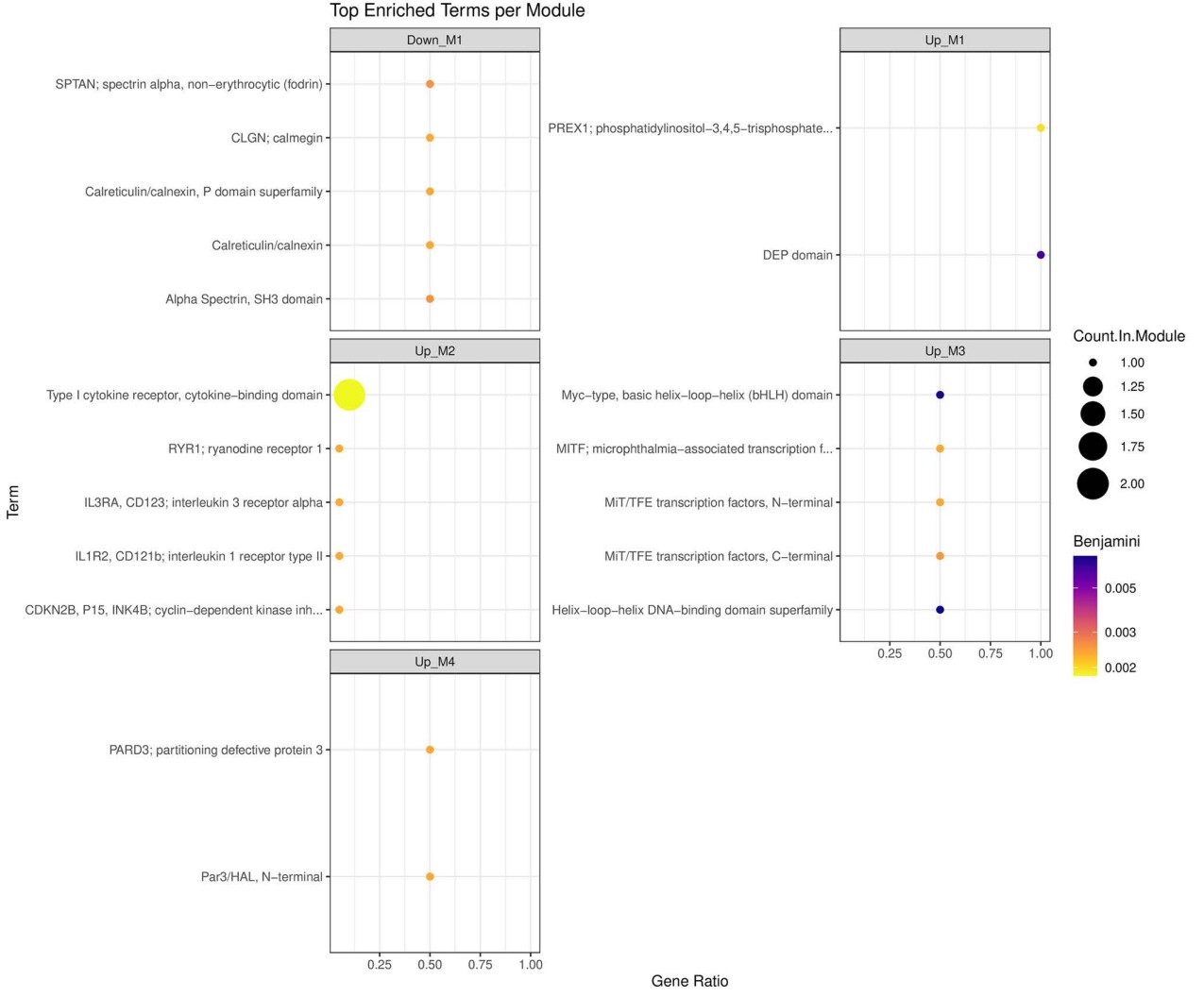

**Fig 5. Summary dot plot of top enriched functional terms for *Healthy* vs *Chronic*.**

calves revealed significant immunomodulatory changes related to processes such as chemotaxis, cytokine-mediated signaling, stress response, and inflammation.

The upregulated modules highlighted key molecular drivers underlying pulmonary inflammation and clinical symptoms such as chemokines. Chemokines exist as a small family of secreted proteins responsible for the stimulation of cellular migration for both homeostatic and inflammatory processes [69–71]. The chemokine system must balance the expression response of proinflammatory and anti-inflammatory cytokines [72–74]. Failure of the host to produce sufficient anti-inflammatories can facilitate chemokine dysregulation, leading to off-target effects (e.g., autoimmune disorders, neoplasia) or ineffective responses (e.g., chronic morbidity) [74,75]. Chemokine dysregulation has been implicated in chronic lung conditions such as cystic fibrosis due to the overproduction of interleukin-8 (IL-8), and is a primary cause of the exaggerated clinical response (e.g., cytokine storm) observed in several viral diseases such as Severe Acute Respiratory Syndrome (SARS) and influenza [74,76]. The role of chemokine involvement in respiratory disease was initially explored in calves infected with Bovine Respiratory Syncytial Virus (BRSV) to serve as a model to understand the age-dependent

risk, pathophysiologic changes, and immune impacts in children infected with Respiratory Syncytial Virus (RSV) [77,78]. A recent study sought to further examine the concentrations of 15 chemokines and 15 cytokines in the plasma of young feedlot bulls diagnosed with BRD. At the individual animal level, the plasma chemokine and cytokine concentrations could not differentiate between the healthy controls and BRD cases, which aligns with previous work by Akter et al. and Chitko-McGown et al. [79,80]. Additional analyses by the authors resulted in the identification of two cytokines (Interleukin-17A and Interferon-gamma) with prognostic value as well as generation of cluster analysis for cytokine profiles relative to health-related variables [81]. Cluster analysis identified three profile subgroups with distinct cytokine expression patterns and health outcomes, underscoring the variability of immune function dynamics relative to previous health history [81]. Further exploration of this variability at the onset of disease is essential for determining what molecular factors drive appropriate infection clearance versus causing pulmonary damage.

Six DEGs identified in the *Healthy* vs *Onset* group were also identified as having predictive potential via machine learning (Table 1). The first DEG highlighted was an uncharacterized protein (LOC104973586). Although there is no direct literature currently published, the identification of this gene by both statistical and machine learning methods warrants functional follow-up and experimental validation due to its potential biological relevance for the onset of BRD.

Integrin-β5 (ITGB5), is a key player underlying pulmonary inflammation and endothelial damage [82,83]. Integrins control the movement and interactions of leukocytes from circulation into tissues. This mechanistic link supports the trafficking dynamics of immune cell recruitment into pulmonary tissue as observed in the downregulated module two (M2) cluster (S8 Table). Integrin dysregulation has been implicated in respiratory disease studies in both human and veterinary medicine [84–86]. Herrick et al. identified both ITGB5 and cyclin-dependent kinase 4 inhibitor B (CDKN2B) as leading edge genes in post-weaned calves with BRD, highlighting them as opportunities for genetic selection strategies [85].

Two isoforms of adhesion G-protein coupled receptor E1 (ADGRE1) were identified as both significantly expressed and predictive of onset of BRD. Adhesion G-protein coupled receptor E1 is strongly expressed on alveolar macrophages in large animal species and humans [87]. Research into ADGRE1 across mammalian species suggests immune selection pressures likely driven by pathogen interactions [88]. Bovine alveolar macrophages are found to have high surface expression of ADGRE1 [89]. Differences in ADGRE1 expression may provide additional insight into BRD associated phenotypes in terms of which animals are more likely to clear initial infection versus progressing to consolidation.

Interlekin-21 (IL-21) is an important cytokine in both innate and adaptive immune response activity [90]. Interleukin-21 and its receptor Interleukin-21 Receptor (IL-21R) have been implicated in the pathogenesis of pulmonary fibrosis and respiratory disease severity [91–93]. Recent evaluation of IL-21 and IL-21R in knockout murine models for respiratory disease found IL-21R signaling to promote M1 polarization of macrophages [93]. Alveolar macrophages in the M1 phenotype are considered pro-inflammatory. While the M1 alveolar macrophages exhibit antimicrobial activity, strong IL-21R signaling can result in exacerbated inflammation and pulmonary damage [93,94]. The results from Yang et al. provide interesting context to this study's observations of IL-21R being identified as differentially expressed and predictive. Further exploration of IL-21R expression in bovine respiratory models is necessary to elucidating its role in BRD progression and ultimately resolution.

Lastly, CDKN2B, which plays an essential role in cell cycle inhibition and immune regulation was identified as differentially expressed and predictive for both *Healthy* vs *Onset* and *Healthy* vs *Chronic* [95]. The upregulation of CDKN2B in the context of the results of this study, signifies a shift in cellular programming away from primary growth activity, as observed in the *Healthy* vs *Onset* downregulated module one cluster, towards an acute inflammatory response. Dysregulation of CDKN2B has been implicated in tumorigenesis and respiratory disease [96–98]. Decreased expression of CDKN2B has been associated with a promotion of fibrosis in patients diagnosed with idiopathic pulmonary fibrosis [99]. The findings by Scruggs et al. in conjunction with the findings of our study highlight CDKN2B as an important immune marker in respiratory disease.

The progression of BRD into a state of chronicity revealed five gene cluster modules pertaining to innate immune signaling suppression, stress response, cytoskeletal remodeling, and transcriptional reprogramming of leukocytes. Key enriched functions pertaining to innate immune signaling included Interleukin-1 Receptor (IL-1R) and Interleukin-1 type II receptor (IL-1R2). Interleukins are large groups of proteins that play an essential role in the activation, differentiation, pro-liferation, maturation, migration, and adhesion of immune cells [100]. Interleukin-1 inflammation contributes to significant pathology related to infection, inflammation, degenerative disease, and neoplasia, which makes it a potential therapeutic target [101,102]. Interleukin-1 type II receptor, also known as a decoy receptor, plays an important role in regulating the action of pro-inflammatory cytokines and chemokines by acting as a scavenging system to dampen cytokine signaling [103,104]. Research examining IL-1R2 expression and function has demonstrated promise as a potential therapeutic target in reducing inflammation severity and exacerbation of clinical signs [74,105]. However, a recent study provided evidence of a paradoxical role of IL-1R2 in the pathophysiology of sepsis. The IL-1R2 receptor demonstrated anti-inflammatory activity in the early stages of sepsis in which IL-1R2 dysregulation occurs, resulting in an immunosuppres-sive state in later stages [106]. The potential for a dueling role associated with IL-1R2 should be explored in other disease processes such as chronic respiratory disease. Exploration of IL-1R2 expression levels could provide additional insight into the degree of chronicity and level of resolution in calves with chronic lung disease.

Additional observations within the *Healthy* vs *Chronic* comparison found upregulated, enriched functional terms pertain-ing to stress response in both module one (M1) and module two (M2). The detection of serine protease inhibitors (SER-PINS) and carbonic anhydrase points to the host's response to injury related to systemic inflammation and pulmonary damage [107,108]. Chronic inflammation promotes dysregulation and transcriptional reprogramming to control cellular fate and function. Subsequent changes to the cytoskeletal structures can alter immune cell migration leading to ineffective response [109].

Ineffective response can lead to chronicity and poor resolution. The resolution of lung disease is an area often over-looked when considering whether disease resolution equates to health. Although exploration of *Healthy vs Resolved* calves in this did not yield significant differentially expressed genes, matrix-remodeling-associated protein 5 (MXRA5) was identified as a top predictive feature for resolved calves. Matrix-remodeling-associated proteins (MXRAs) are specialized components of the extracellular matrix that play an important role in structural scaffold provision and regulation of cell signaling and repair [110]. Dysregulation of MXRAs have been associated in pathogenesis of chronic pulmonary diseases [111–113]. Underlying mechanisms of MXRA5 specifically in the context of respiratory disease remains poorly described. Additional research exploring the intrinsic drivers behind BRD resolution is required.

This initial exploration of the peripheral leukocyte transcriptome of preweaned Holstein dairy calves across BRD stages provided insight into genes that were not only differentially expressed but also be predictive of BRD. The identification of these genes may serve as the starting point for identifying biomarkers suitable for therapeutic targets, genetic improvements, and diagnostic development. The transcriptional changes discussed in this study not only underscore the complexity of calf BRD but highlight gene features that may drive our understanding of nuanced calf-hood BRD phenotypes.

## Conclusion

Overall, the peripheral leukocyte transcriptomic changes associated with BRD stages encompass a diverse range of cellular functions and immunomodulatory processes. Exploration of highly significant genes associated with each disease state provided insight into the host's homeorhetic regulation. Identification of molecular players such as interleukins, integrins, serine protease inhibitors, and matrix-remodeling-associated proteins provide a starting point for future research into the transcriptomic factors driving differences in disease phenotypes. The integration of omics technology into BRD management may help mitigate the detrimental impacts on calf health and farm produc-tivity. As demonstrated in this study, from the onset of disease preweaned calves begin the process of reallocating

resources toward immune function activities. The downregulation of primary metabolic processes toward immuno-logic activity may underly impacts on growth historically reported among calves diagnosed with respiratory disease. Exploration of these differences within animals at the onset of disease could provide insight into factors driving disease chronicity versus resolution. Understanding the ability of animals to effectively and efficiently respond to disease challenges is key to our effort to identify genetic features that contribute to disease susceptibility and robustness. The ability to select more robust animals can enhance animal well-being through reduction of disease incidence, which increases the sustainability of our food systems and improves antimicrobial stewardship. Future research is needed to 1) explore the identified gene features in diverse environments and management systems, 2) quantify the expression levels of potential therapeutic gene targets, and 3) examine the cumulative impacts of transcriptomic changes associated with chronic preweaned lung consolidation and disease resolution on subsequent health and production metrics in an animal's lifetime.

## Supporting information

**S1 Table. Demographic characteristics of Holstein heifer calves selected for BRD comparisons.**
(DOCX)

**S2 Table. Summary of RNA integrity and purity parameters for peripheral leukocyte transcriptomic samples.**
(DOCX)

**S3 Table. Descriptive summary table of differentially expressed genes (DEGs: n = 163; FDR<0.05 and |logFC|>1) identified for *Healthy* vs *Onset*.**
(DOCX)

**S4 Table. Descriptive summary table of differentially expressed genes (DEGs: n = 27; FDR<0.05 and |logFC|>1) identified for *Healthy* vs *Chronic*.**
(DOCX)

**S5 Table. Descriptive summary table of random forest gene features (n = 51) identified for *Healthy* vs *Onset*.**
(DOCX)

**S6 Table. Descriptive summary table of random forest gene features (n = 38) identified for *Healthy* vs *Chronic*.**
(DOCX)

**S7 Table. Descriptive summary table of random forest gene features (n = 57) identified for *Healthy* vs *Resolved*.**
(DOCX)

**S8 Table. Descriptive summary table of significantly enriched terms for *Healthy* vs *Onset*.**
(DOCX)

**S9 Table. Descriptive summary table of significantly enriched terms for *Healthy* vs *Chronic*.**
(DOCX)

**S1 Fig. RNA sequencing quality histogram.** Each line represents a single 2021 sample. All sample Phred Scores indicate high sequence quality.
(TIF)

**S2 Fig. RNA sequencing quality histogram.** Each line represents a single 2022 sample. All sample Phred Scores indicate high sequence quality.
(TIF)

**S3 Fig. Upset plot for the intersection of differential expressed genes and random forest genes for *Healthy* vs *Onset*.** The vertical bars define the intersection size. The horizontal bars show the size of each set. The filled circles in a column set indicates that the set includes an intersection.
(TIF)

**S4 Fig. Upset plot for the intersection of differential expressed genes and random forest genes for *Healthy* vs *Chronic*.** The vertical bars define the intersection size. The horizontal bars show the size of each set. The filled circles in a column set indicates that the set includes an intersection.
(TIF)

## Acknowledgments

We thank the participating farms and associated personnel for their invaluable assistance with this project.

## Author contributions

**Conceptualization:** Lindsey C. Makratzakis, Neils C. Stegelmeier, Craig S. McConnel.

**Data curation:** Lindsey C. Makratzakis, Joel A. Velasco, Stephen P. Ficklin, Craig S. McConnel.

**Formal analysis:** Lindsey C. Makratzakis, Joel A. Velasco, Stephen P. Ficklin, Craig S. McConnel.

**Funding acquisition:** Stephen P. Ficklin, Craig S. McConnel.

**Investigation:** Lindsey C. Makratzakis, Neils C. Stegelmeier, Holly R. Hinnant, Lily A. Elder, Chris M. Mandella, Rachel A. Leeson, Carolyn D. Fisher, Giovana S. Slanzon, Corinna T. Cauchy, Lindsay M. Parrish, Kathlyn N. Heaton, Craig S. McConnel.

**Methodology:** Lindsey C. Makratzakis, Joel A. Velasco, Stephen P. Ficklin, Corinna T. Cauchy, Lindsay M. Parrish, Craig S. McConnel.

**Project administration:** Craig S. McConnel.

**Resources:** Craig S. McConnel.

**Supervision:** Craig S. McConnel.

**Writing – original draft:** Lindsey C. Makratzakis, Joel A. Velasco, Craig S. McConnel.

**Writing – review & editing:** Stephen P. Ficklin, Corinna T. Cauchy, Lindsay M. Parrish.

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
