## [Decision Letter · Decision Letter 0]

9 Dec 2025

PONE-D-25-57063Peripheral leukocyte transcriptomic changes in preweaned Holstein heifer calves with varying stages of Bovine Respiratory DiseasePLOS One

Dear Dr. Makratzakis,

Thank you for submitting your manuscript to PLOS ONE. After careful consideration, we feel that it has merit but does not fully meet PLOS ONE’s publication criteria as it currently stands. Therefore, we invite you to submit a revised version of the manuscript that addresses the points raised during the review process.

We look forward to receiving your revised manuscript.

Kind regards,

Tofazzal Md Rakib, PhD

Academic Editor

PLOS One

Reviewers' comments:

Reviewer's Responses to Questions

Comments to the Author

1. Is the manuscript technically sound, and do the data support the conclusions?

Reviewer #1: Yes

Reviewer #2: Yes

Reviewer #3: Partly

2. Has the statistical analysis been performed appropriately and rigorously? 

Reviewer #1: Yes

Reviewer #2: Yes

Reviewer #3: Yes

3. Have the authors made all data underlying the findings in their manuscript fully available?

Reviewer #1: Yes

Reviewer #2: Yes

Reviewer #3: No

4. Is the manuscript presented in an intelligible fashion and written in standard English?

Reviewer #1: Yes

Reviewer #2: Yes

Reviewer #3: Yes

5. Review Comments to the Author

Reviewer #1: Bovine Respiratory Disease is a persistent and damaging disease in cattle and efforts to understand the disease in greater detail and potentially identify targets for improving diagnosis and treatment are welcome. The methods are comprehensive and very well written. The manuscript makes a new contribution to our understanding of the disease, it is well written, and the Figures are of high quality.

I have the following comments for the authors:

1. Abstract: More detail needs to be included on the FDR cut off and numbers of DEG for some important contrasts.

2. Table 1 should be included as supplementary data

3. My major issue with this manuscript is the selection of the most appropriate data for inclusion in the Figures. Figure 1 should graphically represent the numbers of DEG in each contrast. Subsequent Figures should show volcan plots for time point comparisons. There is no need for 15 Figures showing functional enrichment visualization of GO terms – the most important ones should be included in the main manuscript, and the others should appear in supplementary data. The cellular component Figures are not as valuable in the context of the aims of the paper. What about a Figure showing the predictive classification of the samples using the RFA conducted? What genes most effectively correctly classified resolvers from non-resolvers?

Reviewer #2: The manuscript by Makratzakis et al. is a very interesting study that aims to identify many leukocyte transcriptomic aspects for the resolution of BRD.

In my opinion, the article is very interesting, but in some aspects it is confusing in its presentation of tables and data. Unfortunately, it is very long and hard to read and the information may appear confusing when reading the manuscript.

I believe that the authors should summarise and reorganise the presentation of the data and their results.

I believe that there is a bias in the study in terms of the number of animals sampled and the detection of active or chronic lesions by ultrasound.

If possible, I would suggest that the authors avoid this division into years and consider only the disease, perhaps including time and stables as variables.

Questions and suggestions:

1) The relevant question that I have is related to Ultrasound scans and how many animals had active or chronic pneumonie per group.

In recent years, many variations have been made to the TUS score (Ollivett and Buczinski, 2016). Recent studies suggest that consolidation can be viewed and interpreted in different ways. Consolidation refers to the functional loss of lung aeration, which is shown in ultrasound in different ways: disappearance of A lines, presence of fluid and B lines (inflammatory phlogosis, probable bacteria? active pneumonia?), lesions and pleural effusions, liver-like lesions (viral lesions or chronic lesions?), or lesions with mixed aspects between these (liver-like associated with B lines). All this is used to identify if the lesion is active or not. (I recommend reading the score by Fiore et al 2022 and modified by Lisuzzo et al., 2024; other authors are Feitoza et al 2024 and 2025.) I am writing you that because I expect a better leukocyte and transcriptome response, especially in active lesions.

- Are the authors able to identify if the lesion is active in the study? Did you have B lines (active)?

- Is it possible to define the differences in leukocyte transcriptomics in active and chronic pneumonia on the basis of these changes?

2) the representation of table 1a, 1b, 1c have to be resumed and clarify for readers

3) I cannot read the tables and I cannot vision figures where you describe relations with genes and other variables. The tables appear cut to the right beyond the ‘Gene Name’ column. Maybe I have techical problem or author had submitted tables in the text in a not correct way.

4) You describe more than 20 tables and figures probably related to the same significance. I ask you to simplify and summarise the results and discussions section.

Reviewer #3: The manuscript presented the molecular mechanisms underlying the Bovine Respiratory Disease in different stage using RNA seq approach although the manuscript provided the interesting findings following concern should be addressed by the authors

1- the introduction and discussion must be flourished by the relevant published papers. for example i found that recently the integration of systems biology approach with machine learning for BRD in cattle"Weighted gene co-expression network analysis identifies functional modules related to bovine respiratory disease" It may better authors discuss and compare their findings with prior reports,

2- The material and method must be provided by details, for example the applied script for machine learning modeling must be provided in supplementary files.

3- for modeling the expression profiles, it may be better author perform the modeling with different ML approach to compare the model with higher accuracy rate

4- authors although mentioned that they perfomed the network construction using PPI, but I have not found any constructed network

5- The must popular approach for ranking and prioritizing the functional genes in network analysis is the hub genes analysis, I dint find such analysis. It is essential that authors perfom and provides the hubs for functional prioritization the candidate genes.

6- Authors must be mentioned that the functional enrichment performed in what level of Gene ontology?

7- It must be perfomed KEGG enrichment analysis for the more dissection of functional enrichment of overlapped genes of DEG and Machine learning candidate genes

Minor revision

1- figure quality is very low

2- it may be appropriate that several Tables transfer to the supplmentary fils, and authors perfom and provides the ven diagram for the coverage analysis of candidate genes beween DEG and ML approach

6. PLOS authors have the option to publish the peer review history of their article (what does this mean?). If published, this will include your full peer review and any attached files.

Do you want your identity to be public for this peer review? For information about this choice, including consent withdrawal, please see our Privacy Policy.

Reviewer #1:  Yes: Kieran Meade

Reviewer #2: No

Reviewer #3: No

You may also use PLOS’s free figure tool, NAAS, to help you prepare publication quality figures: https://journals.plos.org/plosone/s/figures#loc-tools-for-figure-preparation

---

## [Author Response · Author response to Decision Letter 1]

11 Mar 2026

Thank you for taking the time to review our manuscript. A copy of our response to the reviewers letter can be found in the uploaded files. Below is a copy of the response to the reviewers letter.

----

Dear editors and reviewers,

Thank you for taking the time to carefully consider and review our manuscript. Below you will find our responses to the reviewers’ specific comments. Please let me know if you have additional questions or concerns.

Respectfully,

Lindsey Christine Makratzakis

---

Reviewer 1 - Comment 1: Abstract: More detail needs to be included on the FDR cut off and numbers of DEG for some important contrasts.

Author response: Thank you for the suggestion of adding more detail into the abstract. The abstract was reworked to include important details related to FDR cut-off and number of DEGs.

---

Reviewer 1 - Comment 2: Table 1 should be included as supplementary data

Author response: Thank you for your suggestion. Table 1 was moved to the supplementary material.

---

Reviewer 1 - Comment 3: My major issue with this manuscript is the selection of the most appropriate data for inclusion in the Figures. Figure 1 should graphically represent the numbers of DEG in each contrast. Subsequent Figures should show volcan plots for time point comparisons. There is no need for 15 Figures showing functional enrichment visualization of GO terms – the most important ones should be included in the main manuscript, and the others should appear in supplementary data. The cellular component Figures are not as valuable in the context of the aims of the paper. What about a Figure showing the predictive classification of the samples using the RFA conducted? What genes most effectively correctly classified resolvers from non-resolvers?

Author response: Thank you for your feedback regarding narrowing down figures to be included in the main body of the manuscript. Updated figures such as volcano plots have been included. Additionally, the summary figures of functional enrichment pathways have been reduced and added to the supplementary material. Information regarding genes with highest feature importance score can be found in the results and discussion.

---

Reviewer 2 – Comment 1: Reviewer #2: The manuscript by Makratzakis et al. is a very interesting study that aims to identify many leukocyte transcriptomic aspects for the resolution of BRD. In my opinion, the article is very interesting, but in some aspects it is confusing in its presentation of tables and data. Unfortunately, it is very long and hard to read and the information may appear confusing when reading the manuscript.

I believe that the authors should summarise and reorganise the presentation of the data and their results. I believe that there is a bias in the study in terms of the number of animals sampled and the detection of active or chronic lesions by ultrasound.If possible, I would suggest that the authors avoid this division into years and consider only the disease, perhaps including time and stables as variables.

Author Response: Thank you for your thorough feedback and concerns. The authors took the time to remove redundancy and rerun the DEG and ML models by analyzing both years together while controlling for dairy farm and year as a covariate. Additionally, animals enrolled in this study are part of a larger prospective cohort study. Calves were assigned into their respective groups based on clinical assessments and thoracic ultrasound diagnosis of BRD.

---

Reviewer 2 - Comment 2: The relevant question that I have is related to Ultrasound scans and how many animals had active or chronic pneumonie per group.

In recent years, many variations have been made to the TUS score (Ollivett and Buczinski, 2016). Recent studies suggest that consolidation can be viewed and interpreted in different ways. Consolidation refers to the functional loss of lung aeration, which is shown in ultrasound in different ways: disappearance of A lines, presence of fluid and B lines (inflammatory phlogosis, probable bacteria? active pneumonia?), lesions and pleural effusions, liver-like lesions (viral lesions or chronic lesions?), or lesions with mixed aspects between these (liver-like associated with B lines). All this is used to identify if the lesion is active or not. (I recommend reading the score by Fiore et al 2022 and modified by Lisuzzo et al., 2024; other authors are Feitoza et al 2024 and 2025.) I am writing you that because I expect a better leukocyte and transcriptome response, especially in active lesions.

- Are the authors able to identify if the lesion is active in the study? Did you have B lines (active)?

- Is it possible to define the differences in leukocyte transcriptomics in active and chronic pneumonia on the basis of these changes?

Author response: Thank you for your thoughtful insight and feedback. The authors recognize that the expanded scoring systems presented by Fiore et al., 2022 and Lisuzzo et al., 2024 provide increased information regarding the pulmonary pathology appreciable on ultrasonography. However, as highlighted by the review written by Feitoza et al., 2025, operator technique, training, and experience is critical for accuracy and diagnostic consistency. Our authors were focused on identifying the presence of lung consolidation (i.e., yes or no) as defined by Ollivett and Buczinski, 2016 and not on the discrimination of consolidation (e.g., hepatization, fluid alveologram). Since the authors are focused on field research and utilization of thoracic ultrasonography in a practical manner for producers and clinicians, we chose to use the standardized method of calf-side thoracic ultrasonography as defined by Ollivett and Buczinksi, 2016.

---

Reviewer 2 - Comment 3: the representation of table 1a, 1b, 1c have to be resumed and clarify for readers

Author response: Thank you for your input regarding the clarity of table 1. The clarity of Table 1a-1c has been improved and moved to supplemental as suggested by reviewer 1.

---

Reviewer 2 - Comment 4: I cannot read the tables and I cannot vision figures where you describe relations with genes and other variables. The tables appear cut to the right beyond the ‘Gene Name’ column. Maybe I have techical problem or author had submitted tables in the text in a not correct way.

Author response: Thank you for your response regarding table visualization. Per PLOS One table guidelines (https://journals.plos.org/plosone/s/tables), the tables were created and formatted under appropriate guidelines as shown below.

---

Reviewer 2 - Comment 5: You describe more than 20 tables and figures probably related to the same significance. I ask you to simplify and summarise the results and discussions section.

Author response: Thank you for your input regarding consolidating the tables and figures. Figures and tables have been updated and condensed based on the feedback provided by all three reviewers.

---

Reviewer 3 – Comment 1: the introduction and discussion must be flourished by the relevant published papers. for example i found that recently the integration of systems biology approach with machine learning for BRD in cattle "Weighted gene co-expression network analysis identifies functional modules related to bovine respiratory disease" It may better authors discuss and compare their findings with prior reports,

Author response: Thank you for your feedback regarding the introduction and discussion. Additional papers were added to the introduction and discussion to highlight relevant work related to our study findings.

---

Reviewer 3 – Comment 2: The material and method must be provided by details, for example the applied script for machine learning modeling must be provided in supplementary files.

Author response: Thank you for this comment regarding the applied script for the machine learning model. This material is now referenced within the manuscript and can be found: https://gitlab.com/mcconnel-projects/nifa-ideas-disability-weights. The GITLAB link will be made public following publication.

---

Reviewer 3 – Comment 3: for modeling the expression profiles, it may be better author perform the modeling with different ML approach to compare the model with higher accuracy rate

Author response: We appreciate your thoughtful assessment of the chosen ML approach. The primary focus of this manuscript was identification of DEGs in BRD progression states relative to healthy. Utilization of a ML approach was a secondary objective to explore potential genes in common between the two parallel methodologies. The authors recognize that the results may vary depending on the complexity of the ML strategy. Random forest machine learning model was chosen based on stability analysis. Our co-author currently has a paper under review that addresses this topic of stability and accuracy for various ML strategies. The authors felt the random forest approach was the appropriate strategy given the overall goal of this particular study.

---

Reviewer 3 – Comment 4: authors although mentioned that they perfomed the network construction using PPI, but I have not found any constructed network

Author response: Thank you for this point of clarity. The authors utilized STRING DB, which generates PPI networks and network summaries as described and referenced in the dataset. The authors chose to present the summary graphics of the PPI generated by string DB as the authors felt they were easier to contextualize for the reader versus a network node image. However, given joint feedback from the reviewers, the authors decided to remove the STRING DB component to reduce redundancy in the presented results.

---

Reviewer 3 – Comment 5: The must popular approach for ranking and prioritizing the functional genes in network analysis is the hub genes analysis, I dint find such analysis. It is essential that authors perfom and provides the hubs for functional prioritization the candidate genes.

Author response: Thank you for your feedback. The network analysis strategy previously used did not utilize network hub analysis. Additionally, given joint feedback from the reviewers, the authors chose to remove this section from the manuscript to reduce redundancy in the results.

---

Reviewer 3 – Comment 6: Authors must be mentioned that the functional enrichment performed in what level of Gene ontology?

Author response: Thank you for your comment that highlights a point of clarification. The authors described the functional enrichment analysis in lines 360-369. The program FUNC-E, as referenced in the manuscript, performs functional enrichment using the following vocabularies: GO, IPR, and KEGG. All levels of the GO vocabulary (Biological Processes, Cellular Components, and Molecular Function are considered for inclusion. The cited references for FUNC-E go into greater detail.

---

Reviewer 3 – Comment 7: It must be perfomed KEGG enrichment analysis for the more dissection of functional enrichment of overlapped genes of DEG and Machine learning candidate genes

Author response: Thank you for your comment and for highlighting a point of clarification. The KEGG enrichment was performed through the program FUNC-E. Significant KEGG results are listed in the results when applicable.

---

Reviewer 3 – Minor revision comment 1: figure quality is very low

Author response: Thank you for your feedback. All figures were run through the PLOS One software PACE for image quality and were approved. Downloading the TIF file and viewing on desktop can provide increased resolution for viewing the figures.

---

Reviewer 3 – Minor revision comment 2: it may be appropriate that several Tables transfer to the supplmentary fils, and authors perfom and provides the ven diagram for the coverage analysis of candidate genes beween DEG and ML approach

Author response: Thank you for this insight. The tables and figures were consolidated and reorganized to incorporate all feedback from the reviewers. Upset plots were generated to describe the intersection of candidate genes between DEG and ML strategies

---

---

## [Decision Letter · Decision Letter 1]

28 Apr 2026

Peripheral leukocyte transcriptomic changes in preweaned Holstein heifer calves with varying stages of Bovine Respiratory Disease

PONE-D-25-57063R1

Dear Dr. Makratzakis,

We’re pleased to inform you that your manuscript has been judged scientifically suitable for publication and will be formally accepted for publication once it meets all outstanding technical requirements.

Kind regards,

Tofazzal Md Rakib, PhD

Academic Editor

PLOS One

Additional Editor Comments (optional):

Reviewers' comments:

Reviewer's Responses to Questions

Comments to the Author

1. If the authors have adequately addressed your comments raised in a previous round of review and you feel that this manuscript is now acceptable for publication, you may indicate that here to bypass the “Comments to the Author” section, enter your conflict of interest statement in the “Confidential to Editor” section, and submit your "Accept" recommendation.

Reviewer #1: All comments have been addressed

2. Is the manuscript technically sound, and do the data support the conclusions?

Reviewer #1: Yes

3. Has the statistical analysis been performed appropriately and rigorously? 

Reviewer #1: Yes

4. Have the authors made all data underlying the findings in their manuscript fully available?

Reviewer #1: Yes

5. Is the manuscript presented in an intelligible fashion and written in standard English?

Reviewer #1: Yes

6. Review Comments to the Author

Reviewer #1: My comments have been addressed and the manuscript has been improved. One final comment is that volcano plots are much more informative if the top number of DEG are labelled in the diagram.

7. PLOS authors have the option to publish the peer review history of their article (what does this mean?). If published, this will include your full peer review and any attached files.

Do you want your identity to be public for this peer review? For information about this choice, including consent withdrawal, please see our Privacy Policy.

Reviewer #1: No

---

## [Editor Report · Acceptance letter]

PONE-D-25-57063R1

PLOS One

Dear Dr. Makratzakis,

I'm pleased to inform you that your manuscript has been deemed suitable for publication in PLOS One. Congratulations! Your manuscript is now being handed over to our production team.

Kind regards,

on behalf of

Dr. Tofazzal Md Rakib

Academic Editor

PLOS One